# The Discovery of Antibacterial Cembranoids from the Soft Coral *Lobophytum crassum* by DeepSAT Analysis

**DOI:** 10.3390/md23120468

**Published:** 2025-12-06

**Authors:** Bing Wu, Li-Gong Yao, Ming-Zhi Su, Gui-Ge Hou, Song-Wei Li, Yue-Wei Guo

**Affiliations:** 1School of Pharmacy, Binzhou Medical University, Yantai 264003, China; wb_you@163.com; 2Shandong Laboratory of Yantai Drug Discovery, Bohai Rim Advanced Research Institute for Drug Discovery, Yantai 264117, China; yaoligong@simm.ac.cn (L.-G.Y.); smz0310@163.com (M.-Z.S.); 3School of Medicine, Shanghai University, Shanghai 200444, China

**Keywords:** soft coral, *Lobophytum crassum*, cembranoid, DeepSAT, antibacterial activity, antiproliferative activity

## Abstract

Eight previously unreported cembranoids (**1**–**8**), along with four known ones (**9**–**12**), have been isolated and identified from the soft coral *Lobophytum crissum* collected from the South China Sea under the guidance of HSQC-based DeepSAT analysis for targeted isolation. The structures of the new compounds were determined by comprehensive spectroscopic analysis, QM-NMR, TDDFT-ECD calculations, and comparison with reported literature data from analogs. In in vitro bioassays, all the isolated compounds have been screened for their antibacterial and antiproliferative activities.

## 1. Introduction

Soft corals, as marine characteristic invertebrates, have emerged as a hotspot in marine natural product research since the 19th century due to their rich natural products [1]. Many of these marine invertebrates colonize benthic zones of subtropical and tropical latitudes, especially with a notable distribution in the South China Sea. The literature reported that secondary metabolites of soft corals display promising bioactivity profiles, demonstrating significant anti-inflammatory [2], antimicrobial [3], antioxidant, and antiproliferative activity [4] and neuroprotective properties [5].

Genus *Lobophytum*, which was one of the first groups to receive attention and also includes the most researched marine invertebrates, is a pivotal soft coral group in marine natural product research. Most compounds from the soft coral of the genus *Lobophytum* are cembranoids and lobane-type diterpenoids [6,7,8,9]. Beyond these diterpenoids, other types of compounds have been reported, such as steroidal derivatives, sesquiterpenoids, lipid-associated compounds, and architecturally novel diterpenoid variants, collectively underscoring its exceptional chemical diversity.

DeepSAT represents an emerging network-based technology capable of effectively predicting the carbon skeleton, structural class, and molecular weight of small molecules directly from ^1^H–^13^C HSQC spectra [10]. The traditional chemical investigation process, which was characterized by high uncertainty, strong dependence on experimental trials, and low efficiency, has been significantly improved by the DeepSAT analysis approach. By reducing the time and resources wasted on blind isolation steps, this method enhances both the efficiency and precision of targeted compound purification in natural product research [11,12,13,14]. Herein, we reported the application of the DeepSAT analysis-guided isolation, structural elucidation, and biological evaluation of these secondary metabolites from the soft coral *Lobophytum crissum*.

## 2. Results

### 2.1. Utilizing DeepSAT to Localize the Target Compound

The acetone extract of the coral sample was separated using diethyl ether and water to obtain an ether-soluble part. This part was further purified by silica gel column chromatography with a gradient of petroleum ether (PE) and ethyl acetate (EA) (from 70:1 to 1:3) to gain seven fractions (Fr. A to Fr. G). The processed HSQC spectrum of Fr. D was uploaded to the DeepSAT platform. The results indicated that Fr. D mainly contained diterpenoids, accounting for 97.0% of its composition. The platform further predicted that the diterpenoids in Fr. D are primarily cembranoids with both three-membered and five-membered oxygen-containing rings, according to its prioritization ranking and cosine score similarity (Figure 1).

### 2.2. Structural Identification of Compounds ***1***–***12***

The biological sample of *Lobophytum crassum* was extracted with acetone, and the extract was further distributed between ether and water. The obtained ether layer was subjected to repeated column chromatography, including silica gel, Sephadex LH-20 gel, and reversed-phase HPLC, leading to the isolation of **12** compounds (**1**–**12**) (Figure 2). Among them, compounds **9**–**12** were rapidly identified as previously reported known compounds 13-acetoxysarcophytoxide (**9**) [15], (+)-laevigatol B (**10**, ${\left[\alpha \right]}_{D}^{25}$ +3.4 (c 0.2, CH_2_Cl_2_)) [16], (+)-sarcophine (**11**, ${\left[\alpha \right]}_{D}^{25}$ +96 (c 0.2, CHCl_3_)) [17], and 1,15*β*-epoxy-deoxysarcophine (**12**) [18], respectively, by comparing their NMR data and optical rotation values with those reported in the literature.

Compound **1** was obtained as an optical active colorless oil {${\left[\alpha \right]}_{D}^{25}$ +25.7 (c 0.21, MeOH)}. The molecular formula was determined as C_22_H_32_O_4_ based on its HR-ESIMS spectrum, which exhibited a quasi-molecular ion peak at *m*/*z* 383.2192 [M + Na]^+^ (calcd. for C_22_H_32_O_4_Na, 383.2198), corresponding to seven degrees of unsaturation. The IR spectrum indicated the presence of a carbonyl group with a strong absorption band at 1738 cm^−1^. The ^1^H (Table 1), ^13^C (Table 2) NMR data and DEPT spectra of **1** displayed 22 distinct carbon signals and enabled unambiguous assignment of four sp^3^ methyl groups (*δ*_C_ 10.3, 15.3, 18.3, 21.4), seven sp^3^ methylene groups (*δ*_C_ 37.6, 24.1, 31.8, 27.7, 32.8, 25.0, 78.4), one sp^2^ methylene group (*δ*_C_ 111.9), three sp^3^ methine groups (*δ*_C_ 84.5, 60.8, 74.69), one sp^2^ methine group (*δ*_C_ 127.3), one sp^3^ quaternary carbon (*δ*_C_ 60.7), four sp^2^ quaternary carbons (*δ*_C_ 129.6, 139.3, 146.7, 132.4), and one carbonyl carbon (*δ*_C_ 170.5). Based on the above, three double bonds and one ester carbonyl occupied four of the seven degrees of unsaturation, indicating that a tricyclic ring system belonged to the structure of **1**. The comprehensive analysis of the ^1^H–^1^H COSY spectrum revealed the presence of four distinct spin systems: H-5a (*δ*_H_ 2.36)/H-6a (*δ*_H_ 1.75)/H-7 (*δ*_H_ 2.67), H-9a (*δ*_H_ 1.80)/H-10b (*δ*_H_ 1.50)/H-11 (*δ*_H_ 5.13), H-2 (*δ*_H_ 5.43)/H-3 (*δ*_H_ 5.30), and H-13a (*δ*_H_ 2.03)/H-14a (*δ*_H_ 2.14) (Figure 3). Furthermore, the key HMBC correlations from H_3_-18 to C-3, C-4, and C-5; from H_3_-19 to C-7, C-8, and C-9; from H_2_-20 to C-11, C-12, and C-13; from H-16a to C-1; from H_3_-17 to C-1, C-15, and C-16; and from H-11 to carbonyl carbon collectively established the planar structure of **1**.

The NOESY spectrum supported the assignment of *E* geometry to the Δ^3,4^ double bond based on a distinct correlation between H-3 (*δ*_H_ 5.30) and H-5a (*δ*_H_ 2.36). The absence of a distinct NOE correlation between H-7 (*δ*_H_ 2.67) and H_3_-19 (*δ*_H_ 1.25) suggested their spatially opposed orientations. The relative configuration at C-2 could not be unambiguously determined based on NOESY correlations (Figure 4). To establish the complete relative stereochemistry of compound **1**, quantum chemical NMR calculations (GIAO method) were performed on all plausible diastereomers, followed by DP4+ probability analysis of the calculated versus experimental NMR data. The DP4+ analysis assigned a 100% probability to the (2*R**, 7*S**, 8*S**, 11*S**) diastereomers, thereby clearly defining the relative configuration of compound **1**. The absolute configuration of compound **1** was further determined by TDDFT-ECD calculations. As shown in Figure 5, the experimental ECD spectrum of **1** exhibited a positive Cotton effect at 202 nm. This pattern matched well with the calculated curve for (2*S*, 7*R*, 8*R*, 11*R*)-**1** but was completely opposite to that of (2*R*, 7*S*, 8*S*, 11*S*)-**1**. Therefore, the absolute configuration of **1** was determined to be 2*S*, 7*R*, 8*R*, 11*R*, as shown in Figure 1.

Compound **2** was obtained as a colorless oil with a specific rotation of ${\left[\alpha \right]}_{D}^{25}$ +28.0 (c 0.10, MeOH). Its molecular formula was determined to be C_20_H_30_O_4_ based on the quasi-molecular ion peak at *m*/*z* 357.2051 [M + Na]^+^ (calcd. for C_20_H_30_O_4_Na, 357.2042) in the HR-ESIMS spectrum, corresponding to six degrees of unsaturation. The IR spectrum indicated the presence of a hydroxy group, evidenced by a broad absorption at 3310 cm^−1^. The ^13^C NMR and DEPT spectra of **2** exhibited 20 distinct carbon signals, which were categorized as follows: three sp^3^ methyls (*δ*_C_ 10.3, 15.5, 18.6), seven sp^3^ methylenes (*δ*_C_ 37.5, 24.5, 31.5, 26.8, 32.9, 25.4, 78.4), one sp^2^ methylene (*δ*_C_ 112.5), three sp^3^ methines (*δ*_C_ 84.5, 60.0, 87.3), one sp^2^ methine (*δ*_C_ 127.2), one sp^3^ quaternary carbon (*δ*_C_ 60.3), and four sp^2^ quaternary carbons (*δ*_C_ 129.6, 139.4, 147.5, 132.4). A detailed comparison of the NMR data of compounds **1** and **2** revealed a high degree of similarity, with a notable downfield shift observed for C-11 (*δ*_C_ 87.3) in **2** compared to that in compound **1** (*δ*_C_ 74.7). In addition, the ^1^H NMR spectrum of **2** displayed a characteristic singlet at *δ*_H_ 7.96, attributable to a hydroperoxy proton. These observations suggested the presence of a hydroperoxy group at C-11 in **2**, which was further supported by the molecular ion peak and IR spectroscopic data. Finally, the planar structure of **2** was unequivocally established by comprehensive analysis of the ^1^H−^1^H COSY and HMBC spectra (Figure 3). In the NOESY experiment, the key cross-peak from H-3 (*δ*_H_ 5.30) to H-5 (*δ*_H_ 2.35) established the *E* geometry of a Δ^3,4^ double bond. The key correlations were between H-7 (*δ*_H_ 2.71) and H-9b (*δ*_H_ 1.51) and between H_3_-19 (*δ*_H_ 1.27) and H-9a (*δ*_H_ 1.93), indicating that H-7 and H_3_-19 have opposite spatial orientations. Assuming *α*-orientation for H-7, H_3_-19 was therefore assigned *β*-orientation. Together with the outcome of the GIAO/DP4+ calculations, this collectively confirmed the relative configuration of compound **2** to be (2*R**, 7*S**, 8*S**, 11*S**). In this case, the absolute configuration of **2** was also determined by comparing its experimental ECD spectrum with TDDFT-calculated ECD spectra of its two enantiomers. An excellent agreement was observed between the experimental spectrum and the calculated curve for (2*S*, 7*R*, 8*R*, 11*R*)-**2**, particularly evidenced by the positive Cotton effect at 199 nm (Figure 5).

Compound **3** was obtained as a colorless oil with the molecular formula of C_22_H_32_O_4_ (seven degrees of unsaturation) determined by a quasi-molecular ion peak at *m*/*z* 383.2189 [M + Na]^+^ (calcd. for C_22_H_32_O_4_Na, 383.2198) on its HRESI-MS spectrum. The IR spectrum exhibited strong carbonyl absorption at 1744 cm^−1^. Comparative 1D NMR analysis revealed that compound **3** is structurally analogous to the known compound **9**, distinguished by the migration of an acetyloxy group from C-13 in **9** to C-19 in **3**. Analysis of the HMBC spectrum (Figure 3) further validated this conclusion through the following key correlations: from the acetyl methyl protons (*δ*_H_ 2.12, s) to the carbonyl carbon of the acetoxy group (*δ*_C_ 171.0, qC); from both H-19a (*δ*_H_ 4.42, d, *J* = 12.1 Hz) and H-19b (*δ*_H_ 3.94, d, *J* = 12.1 Hz) to the same carbonyl carbon; and from H-19b to C-7 (*δ*_C_ 61.1, CH), C-8 (*δ*_C_ 59.9, qC), and C-9 (*δ*_C_ 34.7, CH_2_). The key NOE correlation from H-3 (*δ*_H_ 5.24) to H-5a (*δ*_H_ 2.33) and from H-11 (*δ*_H_ 5.12) to H-13a (*δ*_H_ 1.93) established the *E* geometry of both Δ^3,4^ and Δ^11,12^ double bonds, respectively. The absence of definitive NOE correlations between H-7 and H_a_-19/H_b_-19 in the NOESY spectrum indicates their opposite spatial orientations. The overall relative configuration of **3** was determined to be (2*R**, 7*S**, 8*R**) by DP4+ probability analysis, and the absolute configuration of **3** was further confirmed by TDDFT-ECD calculations. As shown in Figure 5, the experimental ECD spectrum of **3** displayed a positive Cotton effect at 199 nm. The experimental ECD spectrum of **3** was in good agreement with the TDDFT-calculated ECD spectrum for (2*S*, 7*R*, 8*S*)-**3** but showed an opposite trend to that calculated for (2*R*, 7*S*, 8*R*)-**3**, thus establishing the absolute configuration of **3** as 2*S*, 7*R*, 8*S*.

Compound **4** was also obtained as a colorless oil. Its molecular formula was determined to be C_22_H_32_O_4_ based on HR-ESIMS analysis, which displayed a quasi-molecular ion peak at *m*/*z* 383.2188 [M + Na]^+^ (calcd. for C_22_H_32_O_4_Na, 383.2198). The IR spectrum showed a strong absorption band at 1738 cm^−1^, indicative of a carbonyl group. Compound **4** was found to be structurally analogous to the previously reported known compound **10**. The only structural difference was identified at C-1, where the hydroxyl group in **10** is replaced by an acetoxy group in **4**. This substitution was confirmed by the observation of a significant downfield shift in the chemical shift of C-1 in **4** and was consistent with an observed mass increment of 42 Da, corresponding to the replacement of a hydroxyl group by an acetoxy group. Thus, the planar structure of **4** was determined as shown in Figure 3. Similarly, the NOE correlations from H-3 (*δ*_H_ 5.65) to H-5a (*δ*_H_ 2.42) and from H-10a (*δ*_H_ 2.27) to H_3_-20 (*δ*_H_ 1.60) confirmed the *E* geometries of ∆^3,4^ and ∆^11,12^ double bonds. The key NOE correlations were observed between H_3_-19 (*δ*_H_ 1.29, s) and H-5b (*δ*_H_ 2.30, m) and between H-7 (*δ*_H_ 2.93, dd, *J* = 7.3, 3.8 Hz) and H-5a (*δ*_H_ 2.42, m), which, together with the absence of an NOE relationship between H-7 and H_3_-19, indicated their spatial separation on opposite faces of the molecule. Assuming an *α*-orientation for H-7 and a *β*-orientation for H_3_-19, the relative configuration at C-7 and C-8 is assigned as (7*S**, 8*S**). However, the relative configurations at C-1 and C-2 could not be assigned based on NOESY correlation analysis. Finally, the relative configuration of **4** was assigned to be (1*R**, 2*S**, 7*S**, 8*S**) by QM-NMR calculations. The absolute configuration of **4** was determined by TDDFT-ECD calculations. As shown in Figure 5, the experimental ECD spectrum exhibited a positive Cotton effect at 200 nm, which matched the calculated ECD spectrum for (1*R*, 2*S*, 7*S*, 8*S*)-**4** but was opposite to that of (1*S*, 2*R*, 7*R*, 8*R*)-**4**, indicating that the absolute configuration of **4** as 1*R*, 2*S*, 7*S*, 8*S*.

Compound **5** was obtained as a colorless oil. The molecular formula of **5** was established as C_20_H_28_O_3_ based on the HR-ESIMS ion peak at *m*/*z* 315.1955 [M − H]^−^ (calcd. for C_20_H_27_O_3_, 315.1955), indicating seven degrees of unsaturation. The IR absorptions at 1680 cm^−1^ were due to the ester carbonyl group. The 1D NMR data of **5** (Table 3 and Table 4) showed great similarities to those of the co-occurring compound **11**, indicating that they are structural analogs. In fact, the only difference between these two compounds was that the ^13^C NMR signals for C-4 (*δ*_C_ 141.8) and C-5 (*δ*_C_ 30.4) in **5** are significantly upfield-shifted relative to those in **11**, whereas the signal for C-18 (*δ*_C_ 22.5) is downfield-shifted. These changes were attributed to a different *Z* geometry at the Δ^3,4^ double bond, which was further confirmed by the observation of an obvious NOESY correlation between H-3 (*δ*_H_ 4.89) and H_3_-18 (*δ*_H_ 1.74) (Figure 4). Moreover, the NOESY spectrum revealed a key correlation between H-11 (*δ*_H_ 5.13) and H-13b (*δ*_H_ 1.99), which established the *E* geometry of the Δ^11,12^ double bond. The arrangement of (7*S**, 8*S**)was deduced from the observed NOE correlations between H-7 (*δ*_H_ 2.66, dd, *J* = 6.3, 3.6 Hz) and H-9b (*δ*_H_ 1.28, ddd, *J* = 13.2, 12.5, 5.0 Hz) and between H_3_-19 (*δ*_H_ 1.17, s) and H-9a (*δ*_H_ 2.14, m), along with the absence of a direct NOE correlation between H-7 and H_3_-19. The overall relative configuration of **5** was subsequently established as (2*S**, 7*S**, 8*S**) by QM-NMR calculations. The absolute configuration of **5** was deduced as 2*R*, 7*R*, 8*R* through comparison with the ECD spectroscopic data. There was a positive Cotton effect at 199 nm, which is consistent with the trend of the TDDFT-calculated ECD spectrum of (2*R*, 7*R*, 8*R*)-**5** but opposite to that of (2*S*, 7*S*, 8*S*)-**5** (Figure 6).

Compound **6** was obtained as a colorless oil. Its molecular formula of C_20_H_30_O_4_ was established by the quasi-molecular ion peak at *m*/*z* 333.2078 [M-H]^−^ (calcd. for C_20_H_29_O_4_, 333.2066) based on the HR-ESIMS spectrum, which indicated six degrees of unsaturation. The IR spectrum showed characteristic absorption for a hydroxyl group at 3397 cm^−1^. The presence of a trisubstituted epoxide moiety was indicated by the ^13^C NMR signals for *δ*_C_ 64.7 (C-7, CH) and *δ*_C_ 59.5 (C-8, qC), together with the proton signal of H-7 (*δ*_H_ 2.56, d, *J* = 8.1 Hz) in the ^1^H NMR spectrum. A five-membered ring fused at C-1 and C-2 was inferred based on ^13^C NMR signals for *δ*_C_ 71.3 (C-1, qC) and *δ*_C_ 64.5 (C-15, qC), along with HMBC correlations from H-16a to C-1 and C-15 and from H_3_-17 to C-1, C-15, and C-16 (Figure 3). This fusion also implied the presence of a tetrasubstituted epoxide spanning C-1 and C-15. Comparative analysis with the co-isolated known compound **12**, revealed that compound **6** bears a hydroxy group at C-2 (*δ*_C_ 103.8, qC) and contains a double bond at Δ^4,5^ rather than Δ^3,4^. This conclusion was confirmed by the HMBC correlations from H_3_-18 (*δ*_H_ 1.87, s) to C-3 (*δ*_C_ 36.4, CH_2_), C-4 (*δ*_C_ 132.0, qC), and C-5 (*δ*_C_ 128.5, CH) and was consistent with a mass increment of 16 Da. The NOE correlation between H-5a (*δ*_H_ 5.85) and H_3_-18 (*δ*_H_ 1.87) confirmed the *Z* geometry of the Δ^4,5^ double bond, while the NOE correlation between H-11 (*δ*_H_ 5.07) and H-13a (*δ*_H_ 2.46) determined the geometry of the Δ^11,12^ double bond as *E*. Furthermore, the key NOE correlations were observed between H-7 (*δ*_H_ 2.56, d, *J* = 8.1 Hz) and H-9b (*δ*_H_ 1.12, ddd, *J* = 13.2, 12.7, 3.4 Hz) and between H_3_-19 (*δ*_H_ 1.23, s) and H-9a (*δ*_H_ 2.13, m), which, together with the absence of an obvious NOE correlation between H-7 and H_3_-19, indicated a relative configuration of (7*S**, 8*S**). Eventually, the overall relative configuration of **6** was definitively established as (1*S**, 2*S**, 7*S**, 8*S**, 15*S**) by QM-NMR calculations. As shown in Figure 6, by comparing the experimental ECD spectrum of **6** with the theoretically simulated ECD curve, the positive Cotton effect at 200 nm was consistent with the trend of the calculated ECD spectrum of (1*S*, 2*S*, 7*S*, 8*S*, 15*S*)-**6** but opposite to that of (1*R*, 2*R*, 7*R*, 8*R*, 15*R*)-**6**, which ultimately determined the absolute configuration of **6** as 1*S*, 2*S*, 7*S*, 8*S*, 15*S*.

Compound **7** was obtained as a colorless oil with a specific rotation of ${\left[\alpha \right]}_{D}^{25}$ +32.0 (c 0.10, MeOH). Its molecular formula was determined to be C_21_H_32_O_4_ based on HR-ESIMS protonated molecular ion peak at *m*/*z* 349.2379 [M + H]^+^ (calcd. for C_21_H_33_O_4_, 349.2379), indicating six degrees of unsaturation. The ^1^H and ^13^C NMR data (Table 3 and Table 4) of **7** were extremely similar to those of sarcoboettgerol C, a known cembranoid isolated from the South China Sea soft coral *Sarcophyton boettgeri* [19]. After careful comparison of 1D and 2D NMR spectra, the main differences between **7** and sarcoboettgerol C were the significant downfield shift in the C-14 resonance and the upfield shift in the C-15 resonance of compound **7**, strongly implying that these two compounds shared different configurations at the stereogenic centers C-1 and C-15. In the NOESY experiment, the key cross-peak between H-3 (*δ*_H_ 5.09) and H_3_-18 (*δ*_H_ 1.77) established the *Z* geometry of the Δ^3,4^ double bond, while the key cross-peak between H-11 (*δ*_H_ 5.13) and H-13a (*δ*_H_ 2.67) confirmed the *E* geometry of the Δ^11,12^ double bond. In addition, the NOE correlations of H-7 (*δ*_H_ 2.65) with H-9b (*δ*_H_ 1.26) and of H_3_-19 (*δ*_H_ 1.14) with H-9a (*δ*_H_ 2.11) defined the (7*S**, 8*S**) relative configuration. The overall relative configuration of compound **7** was subsequently established as (1*R**, 2*S**, 7*S**, 8*S**, 15*R**) by DP4+ probability analysis (Appendix A). The absolute configuration of compound **7** was unequivocally assigned by comparing its experimental ECD spectrum with the TDDFT-calculated ECD spectra. As shown in Figure 6, the experimental ECD spectrum exhibited a positive Cotton effect at 200 nm, which matched well with the calculated ECD curve for (1*R*, 2*S*, 7*S*, 8*S*, 15*R*)-**7** but was opposite to that for (1*S*, 2*R*, 7*R*, 8*R*, 15*S*)-**7**. Thus, the absolute configuration of **7** was determined to be 1*R*, 2*S*, 7*S*, 8*S*, 15*R*.

Compound **8** was isolated as an optical active colorless oil with a specific rotation of ${\left[\alpha \right]}_{D}^{25}$ +20.0 (c 0.12, MeOH). The molecular formula of **8** was determined to be C_20_H_32_O_5_ by HR-ESIMS analysis, which displayed a quasi-molecular ion peak at *m*/*z* 375.2139 [M + Na]^+^ (calcd. for C_20_H_32_O_5_Na, 375.2147). The IR spectrum of **8** showed a broad absorption band for the hydroxyl group at 3410 cm^−1^ and a strong absorption band for the carbonyl group at 1759 cm^−1^. We conducted a comparative analysis of the NMR data with those of a reported known compound, secocrassumol, in the literature [20], which revealed a high degree of structural similarity, with the only difference identified at the C-7 position: the known compound features an acetoxy group, whereas compound **8** possesses a hydroxy group. This structural assignment was unambiguously confirmed by the ^1^H−^1^H COSY and key long-range correlations in the HMBC spectrum (Figure 3). In the NOESY experiment, the key cross-peak between H-3 (*δ*_H_ 5.24) and H-5a (*δ*_H_ 2.34, ddd, *J* = 13.6, 10.1, 6.3 Hz) established the *E* geometry of the Δ^3,4^ double bond. Additionally, the key cross-peak between H-7 (*δ*_H_ 3.57) and H_3_-19 (*δ*_H_ 1.35) indicated the relative configuration of C-7 and C-8 to be (7*R**, 8*S**). The relative configuration of compound **8** was finally established as (2*R**, 7*R**, 8*S**, 12*R**) by DP4+ probability analysis, and the absolute configuration of **8** was unequivocally assigned through comparison of the experimental ECD spectrum with the TDDFT-calculated ECD spectra. As shown in Figure 6, the experimental ECD spectrum exhibited a positive Cotton effect at 202 nm, which matched well with the calculated spectrum for (2*S*, 7*S*, 8*R*, 12*S*)-**8** but was opposite to that for (2*R*, 7*R*, 8*S*, 12*R*)-**8**. These results determined the absolute configuration of **8** as 2*S*, 7*S*, 8*R*, 12*S*. We proposed that compound **8** was derived from a cembranoid precursor via oxidative cleavage of the Δ^11,12^ double bond, following lactonization and reduction of ketone carbonyl group. Based on this hypothesis and the biosynthetic pathway of secocrassumol [20], a compound structurally analogous to compound **8**, a plausible biosynthetic pathway leading to compound **8** has been proposed, as shown in Figure 7. In bioactivity tests, compound **8** demonstrated antibacterial effects, which may help corals select beneficial microbial communities, inhibit the overgrowth of harmful bacteria, establish a stable microbial barrier, and enhance their resistance to environmental stress.

### 2.3. In Vitro Biological Assay

All isolated compounds **1**–**12** were evaluated for their antiproliferative activity against MV-4-11, MDA-MB-231, and HT-29 cell lines. Among them, only compound **11** exhibited weak antiproliferative activity against MV-4-11 cells, with an IC_50_ value of 48.82 ± 2.28 mM, whereas compounds **1**–**10** and **12** did not show significant antiproliferative activity against any of the tested cell lines.

Furthermore, all isolated compounds were evaluated for antibacterial activity against *Streptococcus parauberis* KSP28, *Staphylococcus aureus*, *Streptococcus agalactiae*, and *Aeromonas salmonicida*. The results revealed that, compared to the positive control drug tetracycline (MIC value of 3.01 mM), compounds **3**–**8** exhibited moderate antimicrobial effects against *Streptococcus parauberis* KSP28 with MIC values of 36.02, 31.62, 33.42, 34.82, and 35.22 mM, respectively. Comparative analysis of the structures of bioactive compounds **3**–**8** showed that the structural changes in the five-membered ring have little effect on the antibacterial activity. Moreover, comparison the structures of **3**–**8** with inactive compounds **1** and **2** revealed that the Δ^11,12^ double bond is crucial for antibacterial activity, and the displacement of the double bond leads to the loss of antibacterial activity.

## 3. Materials and Methods

### 3.1. General Experimental Procedures

Optical rotations were measured on an Anton Paar MCP 5300 polarimeter (Anton Paar Trading Co., Shanghai, China). IR spectra were recorded on a Nicolet iS50 spectrometer (Thermo Fisher Scientific, Madison, WI, USA). ^1^H and ^13^C NMR spectra were acquired using a Bruker DRX-600 spectrometer (Bruker Biospin AG, Fallanden, Switzerland) with residual CDCl_3_ (*δ*_H_ 7.26 ppm) as the internal reference. High-resolution ESI-MS spectra were obtained on a Zeno TOF 7600 mass spectrometer (SCIEX). Silica gel for column chromatography and TLC silica gel plates were supplied by Qingdao Marine Chemical Co., Ltd. (Qingdao, China). Separation was performed using Sephadex LH-20 gel purchased from Amersham Biosciences (Amersham Biosciences, Uppsala, Sweden). Thin-layer chromatography (TLC) analysis was carried out on pre-coated silica gel plates (Silica Gel 60 GF254, Yantai Zifu Chemical Group Co., Ltd., Yantai, China). Spots on TLC plates were visualized under UV light or by heating after spraying with vanillin-H_2_SO_4_ reagent. Reversed-phase high-performance liquid chromatography (RP-HPLC) was performed on an Agilent 1260 series system equipped with a DAD G1315D detector set at 210 nm (Agilent, Santa Clara, CA, USA). Separation was achieved using an Agilent semi-preparative XDB-C_18_ column (5 μm, 250 × 9.4 mm), while an analytical XDB-C_18_ column (5 μm, 250 × 4.6 mm) was employed for purification. All solvents used for column chromatography and HPLC were of analytical grade (Shanghai Chemical Reagent Co., Ltd., Shanghai, China) or chromatographic grade (Dikma Technologies Inc., Beijing, China).

### 3.2. Animal Materials

The soft coral *Lobophytum crissum* (voucher specimen No.19-XD-16) was collected from Ximao Island, Hainan Province, China, at a depth of −20 m, in June 2019, and identified by Li-Gong Yao from the Shandong Laboratory of Yantai Drug Discovery. The sample is deposited at the Shandong Laboratory of Yantai Drug Discovery.

### 3.3. DeepSAT Analysis

The coral sample was first extracted with acetone and then partitioned between diethyl ether and water to obtain the diethyl ether-soluble extract. This extract was initially separated by silica gel column chromatography, yielding seven fractions (Fr. A–Fr. G).

The HSQC spectra of Fr. D were acquired in CDCl_3_ using tetramethylsilane (*δ*_H_ 7.26/*δ*_C_ 77.16) as an internal standard. After removing interfering signals, the key HSQC spectral data were extracted and saved as comma-separated values (CSV) files, then uploaded to the DeepSAT platform (https://deepsat.ucsd.edu accessed on 31 July 2025) to obtain cosine scores and matched diterpenoid structures [10]. Cosine scores are calculated through mathematical vector analysis, which converts HSQC spectral features into vectors and measures their directional similarity with standard feature vectors of known compounds in the DeepSAT database [10,13]. All resulting cosine scores exceeded 0.8, indicating a strong match between Fr. D and standard diterpenoid features.

### 3.4. Extraction and Isolation

The title organism (after extraction dry weight: 861.7 g) was cut into pieces and subjected to exhaustive acetone extraction (5 × 2 L, 30 min/extraction) at room temperature. After rotary evaporation, the combined organic phase yielded the dark brown crude extract, then underwent sequential extraction with diethyl ether (Et_2_O) and H_2_O (2:1 *v*/*v*, 5 cycles) to afford an Et_2_O-soluble fraction (20.1 g). Primary chromatographic separation was executed on a silica gel column (200~300 mesh), with gradient elution from petroleum ether (PE)/ethyl acetate (EA) 70:1 to 1:3 (*v*/*v*), to obtain seven fractions (Fr. A–Fr. G). Fr. D (933 mg) was chromatographed on the Sephadex LH-20 column eluted with a ternary solvent system of PE/CH_2_Cl_2_/MeOH (2:1:1, *v*/*v*/*v*), yielding eleven subfractions (Fr. DA–Fr. DK). Fr. DH (100.1 mg) was further resolved with silica gel column chromatography (200~300 mesh) using a PE/EA gradient (30:1~2:1, *v*/*v*), generating ten subfractions (Fr. DHA–Fr. DHJ). Fr. DHF (12.7 mg) underwent reversed-phase C-18 chromatography with a MeOH/H_2_O gradient (30~100% MeOH, *v*/*v*), obtaining six subfractions (Fr. DHFA–Fr. DHFF). Fr. DHFB (4 mg) via semi-preparative RP–HPLC (MeOH/H_2_O, 57:43), which yielded compound **6** (1.6 mg, t_R_ = 32.5 min).

Fr. DG (257.4 mg) was subjected to C-18 reversed-phase column chromatography (CH_3_CN/H_2_O, 55:45 *v*/*v*). This separation process acquired seven subfractions (Fr. DGA–Fr. DGG), among which Fr. DGD (136.2 mg) was purified to obtain compound **11**. Fr. DGF (26.8 mg) was chromatographed on a semi-preparative RP-HPLC system (CH_3_CN/H_2_O, 55:45, *v*/*v*), yielding compound **5** (2 mg, t_R_ = 20.5 min). Fr. DGC (8.4 mg) was further purified through semi-preparative RP-HPLC (CH_3_CN/H_2_O, 40:60, *v*/*v*), yielding compound **12** (1 mg, t_R_ = 29.0 min). Fr. DF (97.2 mg) was fractionated by silica gel column chromatography (200~300 mesh) with a gradient elution of PE/EA (50:1~5:1, *v*/*v*), yielding five subfractions (Fr. DFA–Fr. DFE). Fr. DFC (52.1 mg) was further purified via C-18 reversed-phase column chromatography using 30~100% (*v*/*v*) MeOH, resulting in five additional subfractions (Fr. DFCA–Fr. DFCE). Fr. DFCC (15 mg) was subjected to semi-preparative RP-HPLC purification using a mobile phase of MeOH/H_2_O (60:40, *v*/*v*), obtaining compound **9** (2.8 mg, t_R_ = 46.5 min). Fr. DFCD (16.3 mg) was resolved by semi-preparative RP-HPLC employing a mobile phase of MeOH/H_2_O (65:35, *v*/*v*). This purification process yielded three compounds: compound **1** (2.1 mg, t_R_ = 22.0 min), compound **4** (0.6 mg, t_R_ = 26.0 min), and compound **3** (0.8 mg, t_R_ = 40.0 min).

Fr. E (1.7374 g) was separated by the Sephadex LH-20 column eluted with the PE/CH_2_Cl_2_/MeOH (2:1:1, *v*/*v*/*v*) elution system, acquiring three subfractions (Fr. EA–Fr. EC). Through the elution system with a gradient elution of MeOH/H_2_O (30~100% MeOH), Fr. EB (634.4 mg) was subjected to C-18 reversed-phase column chromatography separated, obtaining ten subfractions (Fr. EBA~Fr. EBJ). Fr. EBF (56.6 mg) gained ten subfractions (Fr. EBFA~Fr. EBFJ), using silica gel column chromatography (200~300 mesh) with a PE/EA (25:1~1:1) elution system. Fr. EBFB (10.5 mg) was purified by semi-preparative RP-HPLC with a CH_3_CN/H_2_O (67:33) elution system, and after that, compound **7** was obtained (0.7 mg, t_R_ = 14.5 min). Fr. EC (705.8 mg) was fractionated by silica gel column chromatography (200~300 mesh), using a PE/EA (20:1~2:1) elution system, and twelve subfractions were acquired. Among them, Fr. ECG (90 mg) was further purified by semi-preparative RP-HPLC with a gradient elution of CH_3_CN/H_2_O (40:60) while at the same time acquiring compound **2** (1 mg, t_R_ = 27.5 min). Fr. ECI (28.3 mg) separated compound **10** (2.1 mg, t_R_ = 18.5 min) after purification by semi-preparative RP-HPLC with a CH_3_CN/H_2_O (40:60) elution system.

Fr. F (1.5178 g) was separated by the Sephadex LH-20 column eluted with a PE/CH_2_Cl_2_/MeOH (2:1:1, *v*/*v*/*v*) elution system, acquiring seven subfractions (Fr. FA~Fr. FG). Eight subfractions (Fr. FEA~Fr. FEH) were segregated from Fr. FE (243.2 mg) through silica gel column chromatography (200~300 mesh), using a PE/EA (60:1~20:1) elution system. Fr. FED (72.3 mg) was separated by a C-18 reversed-phase chromatographic column, and five subfractions (Fr. FEDA~Fr. FEDE) were obtained through gradient elution of MeOH/H_2_O (30–75% MeOH). Compound **8** (0.6 mg, t_R_ = 10.5 min) was isolated by analytical HPLC with a CH_3_CN/H_2_O (30:70) elution system.

#### 3.4.1. Compound **1**

Colorless oil, ${\left[\alpha \right]}_{D}^{25}$ +25.714 (c 0.21, MeOH); IR (KBr): ν_max_ 2925, 2855, 1738, 1456, 1456, 1433, 1373, 1238, 1039 cm^−1^; For ^1^H and ^13^C NMR spectroscopic data, see Table 1 and Table 2; HRESIMS *m*/*z* 383.2192 [M + Na]^+^ (calcd. for C_22_H_32_O_4_Na, 383.2198).

#### 3.4.2. Compound **2**

Colorless oil, ${\left[\alpha \right]}_{D}^{25}$ +28.000 (c 0.10, MeOH); IR (KBr): ν_max_ 3310, 2925, 2854, 1748, 1455, 1384, 1037, 920 cm^−1^; For ^1^H and ^13^C NMR spectroscopic data, see Table 1 and Table 2; HRESIMS *m*/*z* 357.2051 [M + Na]^+^ (calcd. for C_20_H_30_O_4_Na, 357.2042).

#### 3.4.3. Compound **3**

Colorless oil, ${\left[\alpha \right]}_{D}^{25}$ +20.0 (c 0.10, MeOH); IR (KBr): ν_max_ 2924, 2852, 1744, 1237, 1039 cm^−1^; For ^1^H and ^13^C NMR spectroscopic data, see Table 1 and Table 2; HRESIMS *m*/*z* 383.2189 [M + Na]^+^ (calcd. for C_22_H_32_O_4_Na, 383.2198).

#### 3.4.4. Compound **4**

Colorless oil, ${\left[\alpha \right]}_{D}^{25}$ +12.0 (c 0.10, MeOH); IR (KBr): ν_max_ 2924, 2853, 1738, 1455, 1368, 1239, 1043 cm^−1^; For ^1^H and ^13^C NMR spectroscopic data, see Table 1 and Table 2; HRESIMS *m*/*z* 383.2188 [M + Na]^+^ (calcd. for C_22_H_32_O_4_Na, 383.2198).

#### 3.4.5. Compound **5**

Colorless oil, ${\left[\alpha \right]}_{D}^{25}$ −9.0 (c 0.20, MeOH); IR (KBr): ν_max_ 2924, 2855, 1753, 1680, 1448, 1383, 1090, 1011 cm^−1^; For ^1^H and ^13^C NMR spectroscopic data, see Table 3 and Table 4; HRESIMS *m*/*z* 315.1955 [M − H]^−^ (calcd. for C_20_H_27_O_3_, 315.1955).

#### 3.4.6. Compound **6**

Colorless oil, ${\left[\alpha \right]}_{D}^{25}$ +1.7 (c 0.20, MeOH); IR (KBr): ν_max_ 3397, 2925, 2854, 1509, 1453, 1380, 1250, 1033 cm^−1^; For ^1^H and ^13^C NMR spectroscopic data, see Table 3 and Table 4; HRESIMS *m*/*z* 333.2078 [M − H]^−^ (calcd. for C_20_H_29_O_4_, 333.2066).

#### 3.4.7. Compound **7**

Colorless oil, ${\left[\alpha \right]}_{D}^{25}$ +32.0 (c 0.10, MeOH); IR (KBr): ν_max_ 2926, 2858, 1455, 1382, 1104, 1029, 984 cm^−1^; For ^1^H and ^13^C NMR spectroscopic data, see Table 3 and Table 4; HRESIMS *m*/*z* 349.2379 [M + H]^+^ (calcd. for C_21_H_33_O_4_, 349.2379).

#### 3.4.8. Compound **8**

Colorless oil, ${\left[\alpha \right]}_{D}^{25}$ +20.0 (c 0.12, MeOH); IR (KBr): ν_max_ 3410, 2924, 1759, 1455, 1075 cm^−1^; For ^1^H and ^13^C NMR spectroscopic data, see Table 3 and Table 4; HRESIMS *m*/*z* 375.2139 [M + Na]^+^ (calcd. for C_20_H_32_O_5_Na, 375.2147).

### 3.5. Calculation Section

Traditional methods for determining relative and absolute configurations, such as NOESY and crystallization, are often limited by signal overlap or limited sample availability. Under such constraints, QM-NMR calculations and TDDFT-ECD calculations offer an effective alternative for establishing molecular configurations.

For QM-NMR calculations, conformational searching was first performed in Macro Model 9.9.233 software [Schrodinger, http://www.schrodinger.com/MacroModel (accessed on 31 July 2025)] via the Monte Carlo Multiple Minimum (MCMM) sampling method, using the OPLS_2005 force field and an energy window threshold of 21 kJ/mol (5.0 kcal/mol). Conformers with a Boltzmann population > 1% were then reoptimized at the B3LYP/6-31G (d,p) level. Subsequently, NMR shielding constants for each reoptimized conformer were calculated using the gauge-independent atomic orbital (GIAO) method [21,22,23,24] at the PCM/mPW1PW91/6-311+G (d,p) level via Gaussian 09 [25]. Finally, Boltzmann-weighted average shielding constants were computed for each stereoisomer and correlated with the experimental NMR data.

For the TDDFT-ECD calculations of the compounds, the same method as used in the QM-NMR computations was applied to obtain conformers with Boltzmann populations exceeding 1% after geometric optimization. The TDDFT-ECD calculations were then performed using Gaussian 09 at the B3LYP/6-311G (d,p) theoretical level with the IEFPCM solvation model. Finally, the calculated ECD spectra were generated using SpecDis 171 software, yielding observable results [26].

### 3.6. Antibacterial Activity Bioassays

Four marine bacterial strains (*Streptococcus parauberis* KSP28, *Staphylococcus aureus*, *Streptococcus agalactiae*, and *Aeromonas salmonicida*) were provided by the National Fisheries Research and Development Institute, Korea. The minimum inhibitory concentrations (MICs) of the compounds were measured using the 0.5 McFarland standard method. Mueller–Hinton II broth (cation-adjusted, BD 212322) was used to grow the bacteria. Stock solutions of the compounds were prepared by dissolving them in DMSO at 2 mM.

All samples were first diluted with culture broth to an initial concentration of 100 μM, then subjected to a series of 1:1 dilution with culture broth to obtain concentrations ranging from 100 μM to 0.24 μM. Each dilution (5 μL) was transferred to a 96-well plate, along with three types of controls: sterile controls (no bacteria), growth controls (culture broth with DMSO but no compounds), and positive controls (culture broth with reference antibiotics). All test wells and growth control wells were inoculated with 95 μL of exponentially growing bacterial suspension (approximately 105 CFU per well). The plates were then incubated at 37 °C for 12 h. The MIC value was defined as the lowest concentration that completely inhibited bacterial growth. All MICS results were evaluated according to the standards recommended by the Clinical and Laboratory Standards Institute (CLSI).

### 3.7. Cytotoxicity Testing

Cells were seeded in 96-well plates at 3 × 10^3^ cells per well and incubated overnight. Subsequently, the cells were exposed to various concentrations of the test compounds or 0.1% DMSO (control) for 72 h at 37 °C. Then, 20 μL of MTT solution (5 mg/mL) was added to each well, and the plates were kept in the dark at 37 °C for 4 h. After removing the supernatant, 100 μL of DMSO was added to dissolve the formed formazan crystals. The absorbance was measured at 570 nm using a microplate reader (EnVision, PerkinElmer, UK). Cell viability was calculated using the formula (OD treated − OD blank)/(OD control − OD blank) × 100%. Doxorubicin (DOX) was used as the positive control.

## 4. Conclusions

In summary, this study employed HSQC-based DeepSAT technology to guide the isolation of cembranoids from the soft coral *Lobophytum crissum*, leading to the identification of eight new compounds along with four known compounds. The relative configurations of the new compounds **1**–**8** were established through QM-NMR calculations, while their absolute configurations were determined using TDDFT-ECD computations. This work not only extends the record of cembrane-type diterpenes but also enriches the chemical diversity of the title species.

In the bioassays, these twelve isolated compounds were evaluated for their antibacterial activity against *Streptococcus parauberis* KSP28, *Staphylococcus aureus*, *Streptococcus agalactiae*, and *Aeromonas salmonicida*, as well as for their antiproliferative effects against the MV-4-11, MDA-MB-231, and HT-29 cell lines. The results revealed that compounds **3**–**8** exhibited antibacterial activity against *Streptococcus parauberis* KSP28. These new findings imply that these isolated compounds could provide structural references for the development of new veterinary antibiotics, which could have potential beneficial effects on animal husbandry and fisheries in the future. In addition, compound **11** showed weak selective cytotoxicity against MV-4-11 cells (IC_50_ = 48.82 ± 2.28 µM), with no inhibitory effects observed on MDA-MB-231 or HT-29 cells.

## Figures and Tables

**Figure 1 marinedrugs-23-00468-f001:**
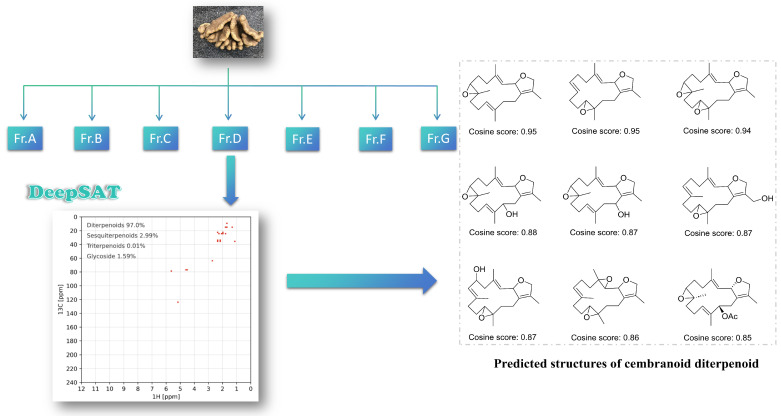
Discovery of cembranoids from the soft coral *Lobophytum crissum* by HSQC-based DeepSAT analysis (taking Fr. D as an illustrative example).

**Figure 2 marinedrugs-23-00468-f002:**
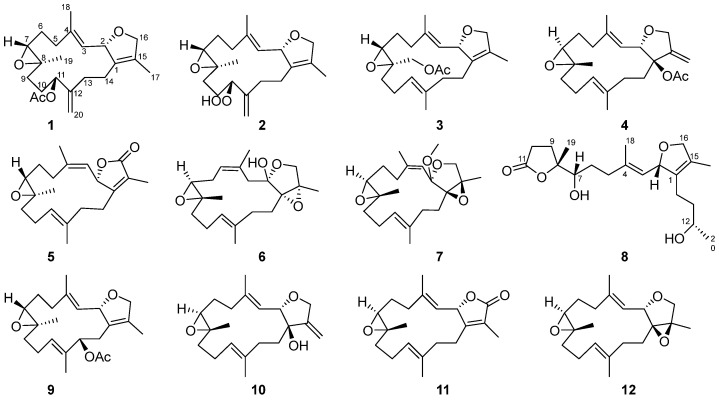
Chemical structures of compounds **1**–**12**.

**Figure 3 marinedrugs-23-00468-f003:**
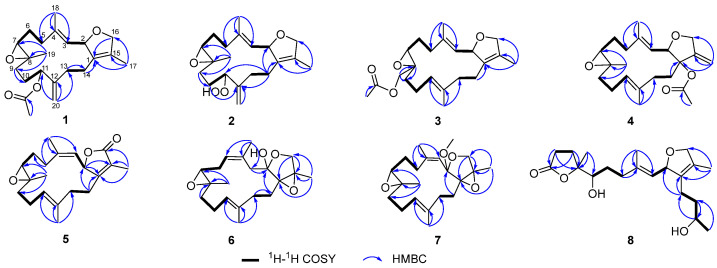
^1^H–^1^H COSY and key HMBC correlations of compounds **1**–**8**.

**Figure 4 marinedrugs-23-00468-f004:**
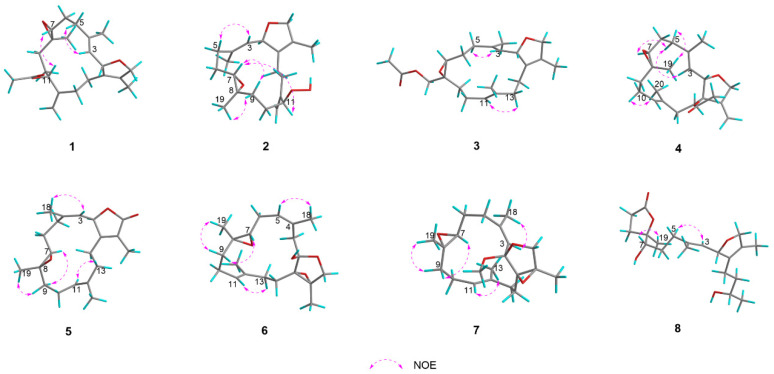
Key NOESY correlations of **1**–**8**.

**Figure 5 marinedrugs-23-00468-f005:**
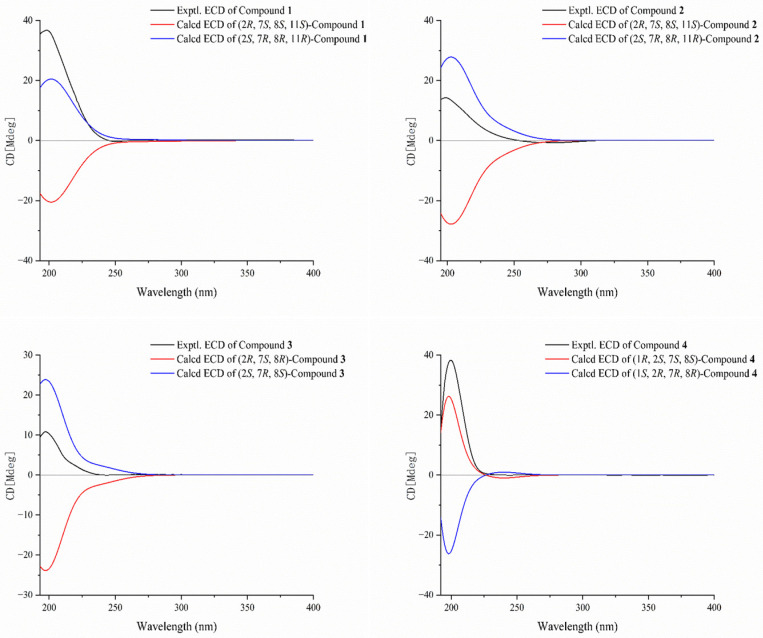
The experimental and calculated ECD spectra for compounds **1**–**4**.

**Figure 6 marinedrugs-23-00468-f006:**
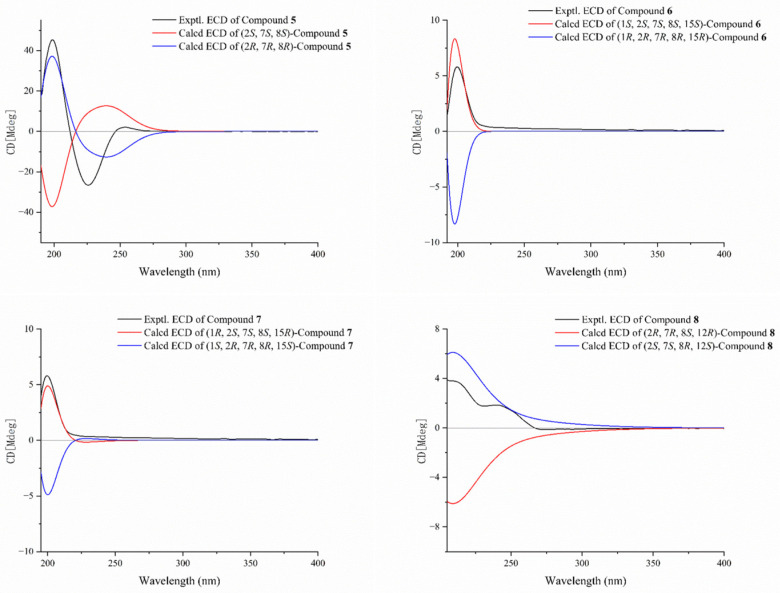
The experimental and calculated ECD spectra for compounds **5**–**8**.

**Figure 7 marinedrugs-23-00468-f007:**
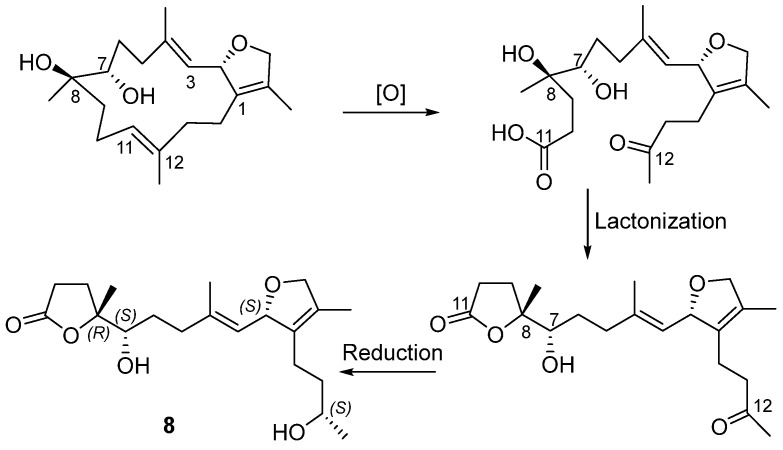
A plausible biosynthetic pathway for the formation of compound **8**.

**Table 1 marinedrugs-23-00468-t001:** ^1^H (600 MHz) NMR data of compounds **1**–**4** in CDCl_3_.

No.	*δ*_H_ Mult. (*J* in Hz)			
	1	2	3	4
2	5.43 m	5.43 m	5.53 m	4.44 d (9.0)
3	5.30 d (9.7)	5.30 d (9.9)	5.24 d (10.0)	5.65 d (9.0)
5a	2.36 m	2.35 m	2.33 m	2.42 m
5b	2.36 m	2.35 m	2.33 m	2.30 m
6a	1.75 m	1.81 m	1.97 m	2.05 m
6b	1.75 m	1.71 td (5.4, 3.9)	1.70 m	1.71 m
7	2.67 m	2.71 t (5.3)	2.87 m	2.93 dd (7.3, 3.8)
9a	1.80 m	1.93 m	2.36 m	2.09 m
9b	1.34 ddd (13.3, 7.7, 5.0)	1.51 m	0.99 td (13.3, 3.2)	1.03 td (13.2, 3.4)
10a	1.70 m	1.56 s	2.25 m	2.27 m
10b	1.50 m	1.39 ddt (14.0, 8.6, 5.9)	1.93 m	1.91 m
11	5.13 dd (7.5, 4.0)	4.24 dd (8.8, 4.0)	5.12 m	5.02 m
13a	2.03 m	2.09 m	1.93 m	2.11 m
13b	2.03 m	1.99 m	1.93 m	2.05 m
14a	2.14 m	2.17 m	2.55 dd (14.1, 11.1, 7.9)	2.47 m
14b	2.14 m	2.09 m	1.70 m	2.09 m
16a	4.55 dd (11.7, 4.5)	4.56 dd (11.9, 5.2)	4.50 m	4.60 m
16b	4.46 dd (11.6, 3.6)	4.46 m	4.50 m	4.32 dt (13.1, 2.3)
17a	1.67 s	1.68 s	1.66 s	5.43 t (2.5)
17b				5.23 t (2.2)
18	1.78 s	1.78 s	1.82 s	1.81 s
19a	1.25 s	1.27 s	4.42 d (12.1)	1.29 s
19b			3.94 d (12.1)	
20a	5.00 s	5.11 dt (21.1, 1.2)	1.60 s	1.60 s
20b	4.95 s			
OAc	2.08 s		2.12 s	1.97 s
-OOH		7.96 s		

**Table 2 marinedrugs-23-00468-t002:** ^13^C (150 MHz) NMR data of compounds **1**–**4** in CDCl_3_.

No.	*δ*_C,_ Type			
	1	2	3	4
1	129.6, qC	129.6, qC	133.2, qC	87.9, qC
2	84.5, CH	84.5, CH	83.8, CH	82.1, CH
3	127.3, CH	127.2, CH	126.8, CH	120.3, CH
4	139.3, qC	139.4, qC	139.0, qC	141.7, qC
5	37.6, CH_2_	37.5, CH_2_	37.8, CH_2_	36.6, CH_2_
6	24.1, CH_2_	24.5, CH_2_	24.8, CH_2_	25.4, CH_2_
7	60.8, CH	60.0, CH	61.1, CH	62.5, CH
8	60.7, qC	60.3, qC	59.9, qC	60.1, qC
9	31.8, CH_2_	31.5, CH_2_	34.7, CH_2_	39.5, CH_2_
10	27.7, CH_2_	26.8, CH_2_	23.1, CH_2_	23.7, CH_2_
11	74.7, CH	87.3, CH	123.7, CH	123.4, CH
12	146.7, qC	147.5, qC	137.4, qC	135.8, qC
13	32.8, CH_2_	32.9, CH_2_	36.8, CH_2_	34.2, CH_2_
14	25.0, CH_2_	25.4, CH_2_	26.2, CH_2_	31.2, CH_2_
15	132.4, qC	132.4, qC	128.2, qC	148.0, qC
16	78.4, CH_2_	78.4, CH_2_	78.6, CH_2_	70.0, CH_2_
17	10.3, CH_3_	10.3, CH_3_	10.4, CH_3_	110.4, CH_2_
18	15.3, CH_3_	15.5, CH_3_	15.8, CH_3_	17.4, CH_3_
19	18.3, CH_3_	18.6, CH_3_	64.8, CH_2_	16.6, CH_3_
20	111.9, CH_2_	112.5, CH_2_	15.3, CH_3_	15.3, CH_3_
OAc	170.5, qC		171.0, qC	170.0, qC
	21.4, CH_3_		21.0, CH_3_	22.5, CH_3_

**Table 3 marinedrugs-23-00468-t003:** ^1^H (600 MHz) NMR data of compounds **5**–**8** in CDCl_3_.

No.	*δ*_H_ Mult. (*J* in Hz)			
	5	6	7	8
2	5.61 d (8.0)			5.40 m
3a	4.89 d (7.5)	3.01 d (13.8)	5.09 m	5.24 dd (9.6, 1.2)
3b		2.27 m		
5a	2.34 m	5.85 dd (11.6, 5.0)	2.62 dd (13.6, 4.0)	2.34 ddd (13.6, 10.1, 6.3)
5b	2.34 m		2.44 dd (13.6, 6.4)	2.27 m
6a	1.91 m	2.64 dd (14.9, 11.4)	1.93 dddd (14.0, 12.8, 6.4, 2.7)	1.62 m
6b	1.48 ddt (14.3, 11.2, 5.7)	1.73 m	1.43 m	1.43 m
7	2.66 dd (6.3, 3.6)	2.56 d (8.1)	2.65 d (2.9)	3.57 dd (11.4, 2.1)
9a	2.14 m	2.13 m	2.11 m	2.66 ddd (18.1, 10.5, 6.4)
9b	1.28 ddd (13.2, 12.5, 5.0)	1.12 ddd (13.2, 12.7, 3.4)	1.26 m	2.57 ddd (18.1, 10.5, 6.6)
10a	2.25 m	1.97 m	2.24 tdd (14.1, 11.2, 4.9)	2.43 ddd (12.9, 10.5, 6.6)
10b	2.14 m	1.85 m	2.13 m	1.79 m
11	5.13 d (10.1)	5.07 d (10.6)	5.13 d (10.9)	
12				3.79 h (6.1)
13a	2.34 m	2.46 ddd (14.0, 8.2, 3.3)	2.67 m	1.49 m
13b	1.99 ddd (14.1, 11.3, 3.3)	2.27 m	2.08 m	1.49 m
14a	2.74 td (12.0, 7.4)	2.27 m	1.81 m	2.02 m
14b	1.87 m	2.08 m	1.81 m	2.02 m
16a		3.82 q (10.0)	3.82 d (9.9)	4.53 dd (11.6, 5.0)
16b		3.82 q (10.0)	3.66 d (9.9)	4.45 ddt (11.7, 2.5, 1.2)
17	1.83 s	1.49 s	1.47 s	1.65 s
18	1.74 s	1.87 s	1.77 d (1.2)	1.76 d (0.8)
19	1.17 s	1.23 s	1.14 s	1.35 s
20	1.68 s	1.64 s	1.61 s	1.19 d (6.1)
2-OMe			3.21 s	

**Table 4 marinedrugs-23-00468-t004:** ^13^C (150 MHz) NMR data of compounds **5**–**8** in CDCl_3_.

No.	*δ*_C,_ Type			
	5	6	7	8
1	162.2, qC	71.3, qC	70.9, qC	133.2, qC
2	79.0, CH	103.8, qC	108.7, qC	84.9, CH
3	121.0, CH	36.4, CH_2_	120.9, CH	127.0, CH
4	141.8, qC	132.0, qC	142.5, qC	139.0, qC
5	30.4, CH_2_	128.5, CH	28.7, CH_2_	35.8, CH_2_
6	26.7, CH_2_	29.2, CH_2_	26.4, CH_2_	27.2, CH_2_
7	64.3, CH	64.7, CH	65.1, CH	74.1, CH
8	60.4, qC	59.5, qC	60.6, qC	89.7, qC
9	37.5, CH_2_	39.0, CH_2_	37.6, CH_2_	30.0, CH_2_
10	24.2, CH_2_	25.9, CH_2_	24.0, CH_2_	27.6, CH_2_
11	125.9, CH	124.6, CH	124.3, CH	178.2, qC
12	134.3, qC	135.7, qC	135.2, qC	68.3, CH
13	38.8, CH_2_	34.9, CH_2_	35.5, CH_2_	37.6, CH_2_
14	25.5, CH_2_	24.4, CH_2_	26.5, CH_2_	21.6, CH_2_
15	122.8, qC	64.5, qC	64.0, qC	128.9, qC
16	174.9, qC	69.8, CH_2_	68.7, CH_2_	78.3, CH_2_
17	8.6, CH_3_	12.7, CH_3_	12.5, CH_3_	10.3, CH_3_
18	22.5, CH_3_	25.0, CH_3_	23.3, CH_3_	15.6, CH_3_
19	16.6 CH_3_	16.2, CH_3_	16.4, CH_3_	23.6, CH_3_
20	15.2, CH_3_	15.5, CH_3_	15.6, CH_3_	24.0, CH_3_
2-OMe			49.4, CH_3_	

## Data Availability

Data are contained within the article or Appendix A.

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
