# Peer review of "The Discovery of Antibacterial Cembranoids from the Soft Coral *Lobophytum crassum* by DeepSAT Analysis"

_marinedrugs, 2025, doi:10.3390/md23120468_

Round 1

Reviewer 1 Report

Comments and Suggestions for Authors

The manuscript presents a comprehensive study on the isolation and structural elucidation of eight new cembranoids from the soft coral Lobophytum crassum, guided by DeepSAT analysis, along with their antibacterial and antiproliferative evaluations. The work is innovative in its application of HSQC-based DeepSAT for targeted isolation and employs advanced spectroscopic and computational methods for structural determination. However, several issues need to be addressed before the manuscript can be considered for publication, including the need for more detailed methodological descriptions, deeper discussion of the biological significance of the results, and clarification of certain structural assignments. Therefore, I recommend major revision.

2. Detailed Comments for the Authors

(1)While the use of DeepSAT for guiding the isolation of cembranoids is a notable strength of this study, the manuscript would benefit from a more detailed explanation of how the DeepSAT platform was specifically applied in this context. For instance, it is not entirely clear how the Cosine score similarity and prioritization ranking were used to predict the presence of cembranoids with specific ring systems. Providing more specifics—such as the key HSQC signals that led to these predictions, the threshold values used for confidence levels, or a comparison with traditional isolation methods—would help readers better understand the advantages and limitations of this approach. Additionally, including a brief discussion on the general applicability of DeepSAT in natural product research would enhance the impact of this methodological innovation and provide a clearer rationale for its use in this study.

(2)The antibacterial screening results indicate that compounds 3–8 exhibit moderate activity against Streptococcus parauberis, which is a valuable finding. However, the discussion of these results is somewhat superficial and lacks a critical analysis of their biological relevance. For example, the authors should compare the MIC values of these compounds with those of other known antibacterial natural products or synthetic drugs to better contextualize their potency. Moreover, the manuscript does not explore the potential mechanism of action or structure-activity relationships (SAR) among the active compounds. Given that some compounds share structural features but differ in activity, a preliminary SAR discussion—even if speculative—could provide useful insights for future drug discovery efforts. Expanding this section would significantly strengthen the biological relevance of the study.

(3)The structural characterization of the new compounds is generally well-supported by NMR, MS, IR, and computational methods. However, the reliance on DP4+ and TDDFT-ECD calculations for stereochemical assignments, while rigorous, is not sufficiently contextualized for non-specialist readers. The authors should briefly explain why these methods were chosen over alternatives, and how the calculated spectra were validated against experimental data. Additionally, for some compounds (e.g., compound 4), the relative configuration at certain centers could not be determined by NOESY, and the DP4+ results are presented without clear interpretation. Providing more explicit reasoning for the final stereochemical assignments—such as highlighting key NOE correlations or contrasting calculated vs. experimental ECD curves—would improve clarity and confidence in the structural conclusions.

(4)The proposed biosynthetic pathway for compound 8 is a thoughtful addition, but it is presented without sufficient supporting evidence or references to analogous transformations in related cembranoids. The authors should either provide preliminary experimental data (e.g., feeding studies or enzyme assays) or cite literature examples where similar pathways have been demonstrated. Furthermore, a brief discussion on the ecological role of these compounds—such as their potential function in coral defense against pathogens or competitors—would add biological context and appeal to a broader readership. This would help bridge the gap between chemical structure and biological function, enriching the narrative of the study.

Comments on the Quality of English Language

The English language used in the manuscript is generally clear and understandable, but there are several instances of awkward phrasing, inconsistent terminology, and minor grammatical errors that detract from the overall readability. For example, in the introduction, the phrase "architecturally novel diterpenoid variants" could be rephrased for better clarity. Additionally, some sentences are overly long and complex, which may confuse readers. I recommend a thorough language polishing by a native English speaker or a professional editing service to improve fluency and consistency.

Author Response

Response to reviewer #1:

The manuscript presents a comprehensive study on the isolation and structural elucidation of eight new cembranoids from the soft coral Lobophytum crassum, guided by DeepSAT analysis, along with their antibacterial and antiproliferative evaluations. The work is innovative in its application of HSQC-based DeepSAT for targeted isolation and employs advanced spectroscopic and computational methods for structural determination. However, several issues need to be addressed before the manuscript can be considered for publication, including the need for more detailed methodological descriptions, deeper discussion of the biological significance of the results, and clarification of certain structural assignments. Therefore, I recommend major revision.

Response: We deeply appreciate the reviewer’s positive comments. The corrections made are listed here below in connection with your concerns:

  1. Detailed Comments for the Authors

(1)While the use of DeepSAT for guiding the isolation of cembranoids is a notable strength of this study, the manuscript would benefit from a more detailed explanation of how the DeepSAT platform was specifically applied in this context. For instance, it is not entirely clear how the Cosine score similarity and prioritization ranking were used to predict the presence of cembranoids with specific ring systems. Providing more specifics—such as the key HSQC signals that led to these predictions, the threshold values used for confidence levels, or a comparison with traditional isolation methods—would help readers better understand the advantages and limitations of this approach. Additionally, including a brief discussion on the general applicability of DeepSAT in natural product research would enhance the impact of this methodological innovation and provide a clearer rationale for its use in this study.

Response: Thanks for your valuable comment. In the revised manuscript, we have added a detailed description of the DeepSAT working procedure and underlying principles in Section 3.3 (Page 11, lines 329–335). In addition, a summary of the limitations of traditional separation methods has been included in the Introduction (Page 1, lines 38–40) to better highlight the advantages of the DeepSAT approach.

(2)The antibacterial screening results indicate that compounds 38 exhibit moderate activity against Streptococcus parauberis, which is a valuable finding. However, the discussion of these results is somewhat superficial and lacks a critical analysis of their biological relevance. For example, the authors should compare the MIC values of these compounds with those of other known antibacterial natural products or synthetic drugs to better contextualize their potency. Moreover, the manuscript does not explore the potential mechanism of action or structure-activity relationships (SAR) among the active compounds. Given that some compounds share structural features but differ in activity, a preliminary SAR discussion—even if speculative—could provide useful insights for future drug discovery efforts. Expanding this section would significantly strengthen the biological relevance of the study.

Response: Thanks for your kind suggestion. A preliminary SAR analysis has been added to the revised manuscript according to the reviewer’s suggestion.

(3)The structural characterization of the new compounds is generally well-supported by NMR, MS, IR, and computational methods. However, the reliance on DP4+ and TDDFT-ECD calculations for stereochemical assignments, while rigorous, is not sufficiently contextualized for non-specialist readers. The authors should briefly explain why these methods were chosen over alternatives, and how the calculated spectra were validated against experimental data. Additionally, for some compounds (e.g., compound 4), the relative configuration at certain centers could not be determined by NOESY, and the DP4+ results are presented without clear interpretation. Providing more explicit reasoning for the final stereochemical assignments—such as highlighting key NOE correlations or contrasting calculated vs. experimental ECD curves—would improve clarity and confidence in the structural conclusions.

Response: Thanks for your kind suggestion. We have added a detailed description of the applicable scenarios and specific procedures for DP4+ and ECD calculations in Section 3.5 (Page 13, lines 424-427). In the original manuscript, the assignment of the (7S*, 8S*) configuration at C-7 and C-8 of compound 4 was based on NOESY analysis implied the spatially opposed orientation of H-7 and H3-19. We have now clarified the underlying structural assumption by explicitly stating the α-orientation of H-7 and β-orientation of H3-19 in the revised text (Page 7, lines 175–176), providing a more complete rationale for the configurational assignment. With the relative configuration at C-7 and C-8 established as (7S*, 8S*), DP4+ analysis was performed to evaluate the four possible stereoisomers of compound 4. The calculated probabilities for the relative configurations were 0% for (1S*, 2R*, 7S*, 8S*), 0% for (1S*, 2S*, 7S*, 8S*), 0% for (1R*, 2R*, 7S*, 8S*), and 100% for (1R*, 2S*, 7S*, 8S*), unambiguously assigning the complete relative configuration as (1R*, 2S*, 7S*, 8S*). Based on the relative configuration (1R, 2S, 7S, 8S) determined by DP4+ analysis, we tentatively assigned the absolute configuration as (1R, 2S, 7S, 8S) for ECD calculations. The resulting calculated ECD curve (red) matches well with the experimental spectrum, confirming the absolute configuration of compound 4 as (1R, 2S, 7S, 8S).

(4)The proposed biosynthetic pathway for compound 8 is a thoughtful addition, but it is presented without sufficient supporting evidence or references to analogous transformations in related cembranoids. The authors should either provide preliminary experimental data (e.g., feeding studies or enzyme assays) or cite literature examples where similar pathways have been demonstrated. Furthermore, a brief discussion on the ecological role of these compounds—such as their potential function in coral defense against pathogens or competitors—would add biological context and appeal to a broader readership. This would help bridge the gap between chemical structure and biological function, enriching the narrative of the study.

Response: Thanks for your kind suggestion. We have added a reference for the biosynthetic pathway of compound 8 (Page 10, lines 278-280), and included a discussion of its potential ecological role in corals (Page 10, lines 281-284).

Reviewer 2 Report

Comments and Suggestions for Authors

The manuscript presents the isolation, characterization and biological studies on novel cembranoids from the soft coral Lobophytum crissum using DeepSAT analysis. The findings in this manuscript are of interest and would be valuable to the relevant research field. The manuscript is recommended for publication in Marine Drugs after the following issues are addressed:

  1. In section 2.1, the manuscript claimed that “Fr. D-F primarily contained diterpenoids (97.0%)”. It’s unclear that whether all three fractions have a of diterpenoid content of 97.0%. According to the isolation results, most of the cembranoids are isolated from fraction D. Are the results fully consistent with the DeepSAT analysis results?

In figure 1, it’s unclear the how the cosine scores were calculated. It’s unclear the score stands for.

  1. Regarding the absolute configurations of compounds 1-3 in section 2.2, the ECD calculation results are not sound convincing, as the (2R, 7S, 8S, 11S)-1 exhibits the completely opposite cotton effect to (2R, 7S, 8S, 11S)-2

Author Response

Response to reviewer #2:

The manuscript presents the isolation, characterization and biological studies on novel cembranoids from the soft coral Lobophytum crissum using DeepSAT analysis. The findings in this manuscript are of interest and would be valuable to the relevant research field. The manuscript is recommended for publication in Marine Drugs after the following issues are addressed:

In section 2.1, the manuscript claimed that “Fr. D-F primarily contained diterpenoids (97.0%)”. It’s unclear that whether all three fractions have a of diterpenoid content of 97.0%. According to the isolation results, most of the cembranoids are isolated from fraction D. Are the results fully consistent with the DeepSAT analysis results?

Response: We are grateful for your careful reading and valuable feedback. The twelve compounds described in this study were isolated from Fr. D to Fr. F. It should be clarified that only the analysis of Fr. D was assisted by the DeepSAT platform, which indicated that "Fr. D mainly contains diterpenoids (97%)." An inaccurate description related to this point has been corrected in the revised manuscript (Page 2, lines 50-55) to prevent any potential misunderstanding.

In figure 1, it’s unclear the how the cosine scores were calculated. It’s unclear the score stands for.

Response: Thanks for your suggestion. After uploading HSQC spectral data to the DeepSAT platform, the system automatically generates cosine scores and matches potential compound structures. The calculation of cosine scores is based on mathematical vector analysis, which essentially converts spectral features of the uploaded HSQC data into vectors and quantifies their directional similarity with standard feature vectors of known compounds in the DeepSAT database. In this manuscript, the cosine scores reflect the directional similarity between Fr. D and the characteristic vectors of diterpenoids. All obtained cosine scores exceeded 0.8, indicating a strong match between Fr. D and the standard features of diterpenoids. A detailed explanation has also been provided on page 11 (lines 329-335) of the revised manuscript.

Regarding the absolute configurations of compounds 1-3 in section 2.2, the ECD calculation results are not sound convincing, as the (2R, 7S, 8S, 11S)-1 exhibits the completely opposite cotton effect to (2R, 7S, 8S, 11S)-2

Response: Thanks for your suggestion. After rechecking and analyzing the detailed ECD calculation process of compounds 1 and 2 carefully, we found that there was no error in the ECD calculation itself for these two compounds. However, a text labeling mistake occurred during the ECD graphing process. For compound 2, the relative configuration (2R*, 7S*, 8S*, 11S*) from the DP4+ calculation results was used as the absolute configuration (2R, 7S, 8S, 11S) to plot the ECD calculated curve (i.e., the red ECD calculated curve), and the result showed that it had an opposite trend to the experimental ECD curve. Unfortunately, due to our carelessness, the red ECD calculated curve (2R, 7S, 8S, 11S) intended for comparison with the experimental ECD curve was incorrectly labeled on the blue ECD calculated curve. This led to an error in the ECD figure of compound 2, which seriously affected the intuitive judgment of its absolute configuration. We have revised Figure 5 and corrected the absolute configuration in lines 137-138 of page 6. We sincerely apologize for this mistake and would like to thank you again for discovering and bringing this issue to our attention.

Reviewer 3 Report

Comments and Suggestions for Authors

The paper is a further report on the chemodiversity of cembrane derivatives from soft corals. The study resulted in the isolation of 8 new and four known cembrane derivative. The “new” compounds are closely related to previous reported analogues. Some of them displayed a moderate  antimicrobial effects against Streptococcus parauberis. The determination of the planar structures is well validated by the experimental data that are discussed in logical way in the paper. The definition of the relative stereochemistry within the cembrane core was based on the analysis of the dipolar couplings by NOESY spectra and QM-NMR calculations on possible stereoisomers, whereas the absolute configuration was assessed by TDDFT-ECD calculations.

It is not up to me to question the reported data, but I am somewhat perplexed to note that the TDDFT-ECD analysis predicts that two compounds  (i.e 1 and 2) almost identical from the point of view of their planar structure and degree of functionalization exhibit the same experimental ECD (and even the same value of optical rotation) despite having an absolute antipodal configuration. Perhaps a chromophore such as a double bond that absorbs in a region close to far UV does not lend itself to a dichroic analysis  I suggest the authors to careful check the ECD calculations.

Compound 7 is a stereoisomer of previously reported sarcoboettgerol C. The NMR spectra were very similar except for some shifting of carbons around the five-member ring, in particular C-14 that represents the most affected nucleus. In my opinion, the proposed stereoisomerism of C1-C15 epoxy ring doesn’t appear the only hypothesis, since the observed effects could be ascribed also to a configurational change around C2.Plase This aspect should be investigated in more detail.

Some minor corrections.

Pag 2 line 64: since the identity of compounds 10 and 11 was also secured by optical rotation data please add also comparison of optical rotation data in the sentence

Pags 12-13 lines 380-412 the digit unit of IR is n, instead of l

Author Response

Response to reviewer #3:

The paper is a further report on the chemodiversity of cembrane derivatives from soft corals. The study resulted in the isolation of 8 new and four known cembrane derivative. The “new” compounds are closely related to previous reported analogues. Some of them displayed a moderate antimicrobial effects against Streptococcus parauberis. The determination of the planar structures is well validated by the experimental data that are discussed in logical way in the paper. The definition of the relative stereochemistry within the cembrane core was based on the analysis of the dipolar couplings by NOESY spectra and QM-NMR calculations on possible stereoisomers, whereas the absolute configuration was assessed by TDDFT-ECD calculations.

Response: Many thanks to your positive comments and valuable suggestions. The manuscript was carefully corrected following your kind instructions and suggestions. The corrections made are listed, point by point, here below in connection with your concerns:

It is not up to me to question the reported data, but I am somewhat perplexed to note that the TDDFT-ECD analysis predicts that two compounds (i.e 1 and 2) almost identical from the point of view of their planar structure and degree of functionalization exhibit the same experimental ECD (and even the same value of optical rotation) despite having an absolute antipodal configuration. Perhaps a chromophore such as a double bond that absorbs in a region close to far UV does not lend itself to a dichroic analysis, I suggest the authors to careful check the ECD calculations.

Response: Thanks for your kind suggestion. After rechecking and analyzing the detailed ECD calculation process of compounds 1 and 2 carefully, we found that there was no error in the ECD calculation itself for these two compounds. However, a text labeling mistake occurred during the ECD graphing process. For compound 2, the relative configuration (2R*, 7S*, 8S*, 11S*) from the DP4+ calculation results was used as the absolute configuration (2R, 7S, 8S, 11S) to plot the ECD calculated curve (i.e., the red ECD calculated curve), and the result showed that it had an opposite trend to the experimental ECD curve. Unfortunately, due to our carelessness, the red ECD calculated curve (2R, 7S, 8S, 11S) intended for comparison with the experimental ECD curve was incorrectly labeled on the blue ECD calculated curve. This led to an error in the ECD figure of compound 2, which seriously affected the intuitive judgment of its absolute configuration. We have revised Figure 5 and corrected the absolute configuration in lines 137-138 of page 6. We sincerely apologize for this mistake and would like to thank you again for discovering and bringing this issue to our attention.

Compound 7 is a stereoisomer of previously reported sarcoboettgerol C. The NMR spectra were very similar except for some shifting of carbons around the five-member ring, in particular C-14 that represents the most affected nucleus. In my opinion, the proposed stereoisomerism of C1-C15 epoxy ring doesn’t appear the only hypothesis, since the observed effects could be ascribed also to a configurational change around C2. This aspect should be investigated in more detail.

Response: Thanks for your suggestion. Compound 7 has the same Z configuration at Δ3,4 as the known compound sarcoboettgerol C, and this conclusion can be drawn from the correlation signals of H-3 (δH 5.09) and H3-18 (δH 1.77) in the NOESY spectrum. Compared with sarcoboettgerol C, we believe that C-14, which shows the largest change in ¹³C NMR chemical shifts among all carbons in compound 7, is slightly farther in space from C-2 and thus less affected by it. In contrast, C-1 is closer to C-14, so the change in the chemical shift of C-14 is most likely caused by the configuration change of C-1. This inference has been confirmed with the help of DP4+ calculations (Isomer 1: 1S*, 2R*, 7S*, 8S*, 15S*; Isomer 2: 1S*, 2S*, 7S*, 8S*, 15S*; Isomer 3: 1R*, 2R*, 7S*, 8S*, 15R*; Isomer 4: 1R*, 2S*, 7S*, 8S*, 15R*) and ECD calculations. Meanwhile, when checking the description of compound 7 in the original manuscript, we found an error in configuration description in lines 250-255 of page 9, which has now been corrected.

Some minor corrections.

Pag 2 line 64: since the identity of compounds 10 and 11 was also secured by optical rotation data please add also comparison of optical rotation data in the sentence

Response: Thanks for your kind suggestion. Compounds 10 ([α]25D +3.4 (c 0.2, CH2Cl2)) and 11 ([α]25D +96 (c 0.2, CHCl3)) were identified by comparing their 1D NMR data and optical rotation values with those of the known compounds (+)-laevigatol B ([α]25D +7.7 (c 1.0, CH2Cl2)) and (+)-sarcophine ([α]25D +92 (c 1.0, CHCl3)), respectively.

Pags 12-13 lines 380-412 the digit unit of IR is n, instead of l

Response: Thank you for pointing out the issue with the IR digit unit. We have revised pages 12-13, lines 392-420 accordingly.

Round 2

Reviewer 1 Report

Comments and Suggestions for Authors

The revised version now presents a clearer and more comprehensive account of the research, particularly regarding the application of the DeepSAT-guided isolation, the structural elucidation methodologies, and the biological activity results. All modifications are appropriate and have effectively enhanced the scientific rigor and readability of the work. Therefore, I think the manuscript meets the journal's standards for acceptance and is now suitable for publication.

Comments on the Quality of English Language

The English language used in the manuscript is generally clear and understandable, but there are several instances of awkward phrasing, inconsistent terminology, and minor grammatical errors that detract from the overall readability. For example, in the introduction, the phrase "architecturally novel diterpenoid variants" could be rephrased for better clarity. Additionally, some sentences are overly long and complex, which may confuse readers. I recommend a thorough language polishing by a native English speaker or a professional editing service to improve fluency and consistency.

Reviewer 3 Report

Comments and Suggestions for Authors

The authors have adequately addressed all raised comments. I am satisfied with the current version, and in my opinion the manuscript can now be accepted for publication.